# Interleukin-13 Mediates Non-Steroidal Anti-Inflammatory-Drug-Induced Small Intestinal Mucosal Injury with Ulceration

**DOI:** 10.3390/ijms241914971

**Published:** 2023-10-07

**Authors:** Rei Kawashima, Shun Tamaki, Yusuke Hara, Tatsunori Maekawa, Fumitaka Kawakami, Takafumi Ichikawa

**Affiliations:** 1Department of Regulation Biochemistry, Kitasato University Graduate School of Medical Sciences, Sagamihara 252-0374, Japan; tamaki.shun@kitasato-u.ac.jp (S.T.); yhara@insti.kitasato-u.ac.jp (Y.H.); maekawa@kitasato-u.ac.jp (T.M.); kawakami@kitasato-u.ac.jp (F.K.); t.ichika@kitasato-u.ac.jp (T.I.); 2Department of Biochemistry, Kitasato University School of Allied Health Sciences, Sagamihara 252-0373, Japan; 3Regenerative Medicine and Cell Design Research Facility, Kitasato University School of Allied Health Sciences, Sagamihara 252-0373, Japan; 4Department of Gastroenterology, Kitasato University School of Medicine, Sagamihara 252-0374, Japan; 5Department of Gastroenterology, Kitasato University Graduate School of Medical Sciences, Sagamihara 252-0374, Japan; 6Department of Health Science, Kitasato University School of Allied Health Sciences, Sagamihara 252-0373, Japan

**Keywords:** IL-13 (Interleukin 13), NSAIDs (non-steroidal anti-inflammatory drugs), small intestine, ulceration, LPMCs (lamina propria mononuclear cells)

## Abstract

Non-steroidal anti-inflammatory drugs (NSAIDs), which are antipyretics and analgesics, cause gastrointestinal disorders, such as inflammation and ulcers. To prescribe NSAIDs more safely, it is important to clarify the mechanism of NSAID-induced gastrointestinal mucosal injury. However, there is a paucity of studies on small intestinal mucosal damage by NSAIDs, and it is currently unknown whether inflammation and ulceration also occur in the small intestine, and whether mediators are involved in the mechanism of injury. Therefore, in this study, we created an animal model in which small intestinal mucosal injury was induced using NSAIDs (indomethacin; IDM). Focusing on the dynamics of immune regulatory factors related to the injury, we aimed to elucidate the pathophysiological mechanism involved. We analyzed the pathological changes in the small intestine, the expression of immunoregulatory factors (cytokines), and identified cytokine secretion and expression cells from isolated lamina propria mononuclear cells (LPMCs). Ulcers were formed in the small intestine by administering IDM. Although the mRNA expression levels of IL-1β, IL-6, and TNFα were decreased on day 7 after IDM administration, IL-13 mRNA levels increased from day 3 after IDM administration and remained high even on day 7. The IL-13 mRNA expression and the secretion of IL-13 were increased in small intestinal LPMCs isolated from the IDM-treated group. In addition, we confirmed that IL-13 was expressed in CD4-positive T cells. These results provided new evidence that IL-13 production from CD4-positive T cells in the lamina propria of the small intestine contributes to NSAID-induced mucosal injury.

## 1. Introduction

Non-steroidal anti-inflammatory drugs (NSAIDs) are widely used for antipyretic and analgesic purposes [1]; however, they are known to cause damage to the gastrointestinal mucosa as a side effect [2,3,4]. Surveys conducted in the United States have shown that approximately 20,000 of the 13 million patients treated with NSAIDs die each year from NSAID-induced gastrointestinal complications [5]. Gastrointestinal bleeding due to NSAIDs used for analgesic purposes is also increasing as a direct cause of death, and it is necessary to pay attention to the careless use of NSAIDs since they are frequently prescribed in clinical settings. It has been reported that the risk of developing peptic ulcers increases when NSAIDs are used in combination with drugs used for the treatment of cardiovascular diseases and osteoporosis. We believe that the elucidation of the mechanism of mucosal injury is an urgent task [6,7,8]. Prostaglandins (PGs), which induce fever and pain due to inflammation, are synthesized using cyclooxygenase (COX) [9]. By inhibiting COX-2, NSAIDs suppress PG production and exhibit anti-inflammatory and analgesic effects [10]. However, NSAIDs inhibit not only COX-2 but also COX-1, which plays a role in gastric mucosal protection, causing mucosal damage [3,4,11,12]. In addition, the production of thromboxane (TXA) is also suppressed [13], which may lead to complications such as bleeding ulcers [14].

NSAID-induced gastric mucosal damage is caused by the inhibition of COX-1-mediated signaling, which in turn inhibits mucosal defense factors. This induces inflammation in the local mucosa, resulting in gastric mucosal damage accompanied by ulceration. Previous studies on NSAID-induced gastric ulcers have reported that the expression of IL-1β, IL-6, and TNF-α increases in the local ulcer mucosa [15,16]. In recent years, advances in endoscopic examination of the lower gastrointestinal tract have revealed that NSAIDs cause mucosal lesions such as ulcers in not only the stomach and duodenum but also in the small intestine and colon [17]. Although the mechanism of injury in the small intestine is expected to be similar to that in the stomach, there may be a mechanism different from that in the stomach because the mucosal environment (parameters such as pH) is different. We have previously reported that NSAID-induced gastrointestinal mucosal injury leads to changes in the intestinal flora [18], and that IL-13 expression is upregulated in radiation-induced enteritis [19,20]. However, there is a paucity of studies on NSAID-induced small intestinal mucosal injury, and it is currently unknown which site of the small intestine suffers inflammation and ulceration and what injury mediators are involved. Therefore, in this study, we created an NSAID-induced small intestinal mucosal injury model, focused on the dynamics of immune regulatory factors related to the injury, and aimed to elucidate the onset mechanism involved.

## 2. Results

### 2.1. Body Weight Change and Effect on Intestinal Tissue

To examine the systemic effects of indomethacin (IDM) administration, body weights were compared before and after indomethacin administration. In the control group, it increased steadily after IDM administration, rising by about 4% compared to the value before IDM administration. In the IDM group, it decreased by about 5% compared to its pre-administration value, and a significant difference was seen between the control group and the IDM group three days after IDM administration (Figure 1a). The food consumption and fecal volume were compared before and after IDM administration, and it was observed that the average food consumption was approximately 3.67 g/mouse in the C group and 2.49 g/mouse in the IDM group, and average fecal volumes were approximately 2.03 g/mouse in the C group and 0.49 g/mouse in IDM groups. These were found to decrease significantly in the IDM group, which is consistent with our previous reports.

The highly elastic gastrointestinal tract is easily affected by edema and fibrosis when injured, causing it to decrease in length. We compared the total length of the small intestine between the control group and the IDM group. It was 34 cm in the IDM group compared to 40 cm in the control group. The IDM group showed a shortening of about 6 cm (about 15%) compared to the control group, showing a sufficiently significant difference (Figure 1b). When the small intestine is injured, the height of the intestinal villi tends to decrease. Comparing villus height in the control and IDM groups on H&E staining images, we found that both of them in the jejunum and ileum decreased by approximately 17% on the seventh day after IDM administration (Figure 1c,d). These results suggest that the administration of IDM potentially damaged the small intestinal mucosa.

### 2.2. Determination of Ulceration

Tissue damage (such as that caused by ulceration) increases vascular permeability; as such, Evans Blue staining is generally used as an index for specifying the ulcer site and size in animal experiments. The jejunum and ileum of the IDM group were highly stained with Evans Blue (Figure 2a). In the histology of the Evans Blue-positive site in the IDM group, ulcers were observed (Figure 2b). When the number of ulcers was counted with the naked eye, both the jejunum and ileum of the control group were negative, standing at about 2.4 in the first half of the jejunum (No. 1), about 3.4 in the latter half of the jejunum (No. 2), about 10.5 in the first half of the ileum (No. 3), and about 7.8 in the latter half of the ileum (No. 4) in the IDM group on the third day after IDM administration. (Figure 2c). In addition, to quantify the severity of ulcers, the amount of extravasated Evans Blue dye in the tissue was eluted from the tissue into formalin and the absorbance was measured. While the absorbance of the control group is all 0.1 or less, it stands at about 0.3 in the first half of the jejunum (No. 1), about 0.5 in the latter half of the jejunum (No. 2), about 0.7 in the first half of the ileum (No. 3), and about 0.8 in the latter half of the ileum (No. 4) in the IDM group on the three days after IDM administration (Figure 2d). After IDM administration, ulcers were observed along the entire length of the small intestine. On the other hand, since the detection rate of Evans Blue tended to be high in the latter half of the small intestine, ulcer formation was thought to occur frequently from the middle part of the small intestine to the ileum.

### 2.3. Gene Expression Analysis of Injury-Related Factors in the Small Intestine

Our analysis of the mRNA expression of several cytokines in the ileum revealed that the mRNA expression of IL-1β, IL-6, and TNFα increased markedly from day 1 to day 3 of IDM administration (Figure 3a). On the other hand, IL-13 mRNA levels increased from day 3 after IDM administration and remained high until even day 7 (Figure 3a). Focusing on the time points at which the mRNA expression of various cytokines was high in Figure 3a, the intestinal tract was analyzed separately by region. IL-1β, IL-6, and TNFα mRNAs were significantly increased, especially in the latter half of the ileum (No. 6–8). Although no changes were observed in IL-4 expression, IL-13 expression was also significantly increased in the latter half of the ileum, and its expression was elevated in all layers (Figure 3b). In other words, it was suggested that IL-13, in addition to IL-1β, IL-6, and TNFα, was involved in IDM-induced mucosal injury.

### 2.4. Relevance of IL-13 for Ulceration

Although IL-13 was expressed in the entire intestinal tract after IDM administration in Figure 3b, we subsequently analyzed its expression in the ulcer, near ulcer, and morphologically preserved part to verify whether it depended on local pathology. IL-13 mRNA expression was analyzed in the small intestinal tissue separated into each site via the laser microdissection method (Figure 4a). IL-13 expression was significantly increased in the near ulcer rather than in the ulcer, and constant expression was confirmed in the morphology-preserving part (Figure 4b). Therefore, it was suggested that IL-13 may contribute to IDM-induced ulceration.

### 2.5. IL-13 Expression in the Small Intestinal Mucosa

Immunohistochemical staining with anti-IL-13 antibody revealed increased IL-13 expression in the lamina propria of the jejunum and ileum after IDM administration (Figure 5a). The IL-13-positive rate per villus increased after IDM administration (Figure 5b). 

### 2.6. Analysis of Lamina Propria Mononuclear Cells (LPMCs)

The mononuclear cell fraction of the lamina propria in the small intestine was separated and the cell numbers were compared. About 0.9 × 10^6^ cells/mouse were obtained in the control group and about 0.4 × 10^6^ cells/mouse were obtained in the IDM group (Figure 6a). The IL-13 mRNA expression levels in LPMCs cultured for 24 h in the presence of PMA and ionomycin for cell activation was significantly increased in the IDM group (Figure 6b). In addition, IL-13 secretion in the supernatant cultured in the presence of PMA and ionomycin was enhanced (Figure 6c). Although the number of LPMCs decreased due to IDM administration, the amount of IL-13 secretion increased, suggesting that stimulation with IDM increased the ability of LPMCs to produce IL-13.

### 2.7. IL-13 Expression of LPMCs

We used flow cytometry to identify the cell subpopulations of small intestinal LPMCs treated with IDM. The abundance ratio of B cells identified using B220 and macrophages identified using F4/80 did not differ significantly between the control group and the IDM group (Figure 7). On the other hand, CD3^+^ CD4^+^ T cells identified using CD3 or CD4 in the IDM group were detected approximately twice as much as in the control group (Figure 7). It was suggested that IDM-induced mucosal injury caused changes in the population of LPMCs. Assuming that IL-13 is produced from significantly increased CD4-positive T cells, we performed FACS analyses via the intracellular staining of IL-13. While the CD4^+^IL-13^+^ T cell population was 12.16% in the control group, it significantly increased to 43.46% in the IDM group (Figure 8a). Furthermore, we performed double staining using CD4 and IL-13 antibodies and confirmed that CD4^+^ T cells tested positive for IL-13 (Figure 8b). Therefore, it was clarified that IL-13 is produced from CD4^+^ T cells in the lamina propria of the small intestine during IDM-induced small intestinal mucosal injury.

### 2.8. Cytotoxicity of IL-13

In concluding that IL-13 is one of the causative agents of IDM-induced gastrointestinal injury, it was necessary to verify whether IL-13 itself is cytotoxic. IL-13 was applied directly to intestinal epithelial cells, and the presence or absence of cell proliferation was evaluated. IL-13 and IDM were added to Caco2, and cell viability was measured in vitro. After the start of the culture, the control group maintained an increase in cell number; however, the cell viability rate decreased by 25% in the IDM-only group (0 μg/mL hIL-13) (red line) (Figure 9). The addition of recombinant IL-13 reduced it further and decreased it by 41%–46% (green and blue lines). Conversely, the suppression of cell proliferation was observed even when recombinant IL-13 was added without IDM (gray line) (Figure 9). Therefore, it was suggested that intestinal epithelial cells may be damaged by IDM and exacerbated by the presence of IL-13.

## 3. Discussion

One of the side effects of NSAIDs is gastrointestinal mucosal injury. There are many reports on gastric mucosal injury, which easily manifests clinically [21,22,23]. However, NSAIDs actually cause tissue damage not only in the stomach but also in the small intestine. Because the small intestine is not easily recognized as a target of treatment in patients presenting with symptoms such as pain, there is a concern that ulcers may worsen, not to mention the ensuing malabsorption of nutrients [24]. In this study, we created a model of NSAID-induced small intestinal mucosal injury and focused on the dynamics of immunoregulatory factors associated with the injury to elucidate the pathogenesis of the phenomenon. We found that IL-13 may be associated with the pathogenesis of mucosal injury in the small intestine.

IDM was selected in the current study because it was found to cause stable intestinal injury in previous studies [25,26,27]. In addition, IDM is mistakenly believed to be safer than aspirin because its current status as a frequently prescribed orthopedic pain medication, such as in the treatment of rheumatism, should mean it has fewer side effects than aspirin; also, it is included in over-the-counter compresses [28,29]. The excessive use of poultices can increase blood levels of IDMs through subcutaneous absorption and cause gastrointestinal disorders. In addition, NSAIDs are often prescribed as oral drugs; however, even when administered orally, they not only act directly on epithelial cells but also traverse a blood-borne route to attack the mucosa. Therefore, the task of creating stable animal models of mucosal injury by IDM, intravenous, intraperitoneal, or subcutaneous injections that are unaffected by the first-pass effect of oral administration, is considered necessary. In this study, intravenous injection was not employed due to the technical problems faced by the experimenter. In addition, although the average dose of NSAIDs administered to the human body is 1 mg/kg per day, we decided to use a dose of 30 mg/kg per day because mice have low drug sensitivity and we wanted to induce acute mucosal injury, and also because this was the dose that captured mucin secretion in previous studies conducted in our laboratory [25,26,27].

First, in the examination of body weight and intestinal length, IDM administration resulted in reductions in body weight and intestinal length (Figure 1). This was because inflammation of the small intestinal mucosa caused its water content to decrease, edema or fibrosis-like transition, and tumor formation, leading to a shortening of the intestinal tract. This could be interpreted as a decrease in weight loss because of decreased nutrient absorption and reduced food intake. The feces of mice treated with IDM exhibited no visible change in color or consistency; however, the water content increased, and pH shifted toward the alkaline range [18]. Although no obvious symptoms were observed, such as diarrhea, we believe that changes in the water content of feces were related to the state of disease.

Vascular permeability at the site of inflammation, as demonstrated by Evans Blue, was predominantly increased in the IDM group. In addition, the incidence of ulceration in the ileum was higher than that in the jejunum (Figure 2). Since the number of ulcers included a factor of superficial, isolated frequency, while the total amount of pigment was intended to examine the severity of ulceration by taking into account the ulcer area and depth, ulcers in the second half of the ileum (Figure 2d-No. 4) were larger in area in the IDM group, suggesting that the ulcers progressed to a greater depth. The results of H&E staining in the IDM group also demonstrated that the ulcers reached the submucosa of the ileum. Therefore, we concluded that, in IDM-induced gastrointestinal injury, ulcers also formed in the small intestine and, in particular, that ulcers were more likely to occur in the ileum. One of the reasons for this is thought to be that the ileum contains many Peyer’s patches (immune organs), and that the ileum is highly sensitive to drugs. In addition, it was thought that ulcers caused by repeated inflammation became more severe in the ileum, which has thinner tissue because they tended to extend deeper.

Next, to examine the immune response in the small intestine, we analyzed cytokine mRNA expression in the small intestine. mRNA expression of IL-1β and IL-6, which were involved in the initial inflammatory response, transiently increased after IDM administration, while IL-13 began to increase after the third day of IDM administration and remained high until the seventh day (Figure 3). This suggested that IL-13 was involved in the chronicity of inflammation. Indeed, IL-13 was often interpreted as an anti-inflammatory cytokine, and it was possible to interpret the increase in IL-13 as a mediator that suppresses inflammation during recovery. However, since the individual observation data such as body weight and histological data showed that the disease continued to deteriorate, we believed that IL-13 was more likely to be interpreted as an injury factor than a healing factor that suppresses inflammation in this animal model. In our previous study, we reported that IL-13 increased in the small intestinal mucosa in radiation-induced small intestinal inflammation and could contribute to mucosal injury [19]. Although IL-4 and IL-13 are said to behave similarly in both allergic reactions, such as respiratory and cutaneous reactions [30,31], only IL-13 alone was upregulated in the present study. This was also the case in our previous study of radiation-induced small intestinal inflammation [19]. As the immune response in the gastrointestinal tract can vary greatly among sites, the tissues were examined on the days when the expression was high by the intestinal site (Figure 3b). Expectedly, the expression levels of all cytokines that were studied increased significantly in the ileum and correlated with ulcer-prevalent sites (Figure 2c,d). On the contrary, a significant difference in IL-13 expression was also observed in the jejunum, suggesting that it is a relatively sensitive mediator in shaping histological pathology. The reason why we could not detect an increase in IL-13 expression in the ulcer area was thought to be that the ulcer area is extremely cytotoxic, resulting in high levels of damage to nucleic acids (Figure 4). On the other hand, IL-13 expression was higher in the vicinity of the ulcer than in the morphological-preserving area (Figure 4), suggesting that IL-13 may contribute to ulcer formation from inflammation.

We expected that the lymphocytes of IDM-induced intestinal mucosal injury model mice were already activated and could produce cytokines; however, sufficient secretion was not confirmed (Figure 5). Although IL-13 was expressed in vivo, its production from isolated cells was not confirmed; thus, we attempted to produce it through stimulation. IL-13 production from lymphocytes isolated from tissues was higher in the IDM group than in the control group. Despite the requirement for further investigation on slowing down of IL-13 production via isolation, it can be argued that the lymphocytes of IDM-induced intestinal mucosal injury model mice are more likely to be activated.

Next, to identify IL-13-producing cells and to confirm IL-13 secretion, LPMCs (cells of the mucosa proper layer) were isolated, and the number of LPMCs decreased with IDM administration. This was thought to be due to IDM-induced tissue disruption, resulting in a decrease in the total number of LPMCs. On the other hand, IL-13 expression and secretion increased despite the decrease in cell number. The FACS results suggested that IL-13 expression was elevated because of the increased number of CD4^+^ T cells, which are IL-13-producing cells (Figure 6 and Figure 7). IL-13 mRNA expression in LPMCs in the IDM group was already elevated before stimulation. This indicates that T cells were already activated via exposure to IDM. However, it is uncertain whether this activation was a direct or indirect effect of IDM. On the other hand, no IL-13 production was observed in cells cultured for 24 h in the non-activated state. This suggests that, although activated, simply culturing the cells did not lead to secretion.

Intracellular staining for IL-13 showed that the CD4^+^ and IL-13^+^ cell population was markedly increased in the IDM group, and immunochemical histological staining confirmed that CD4^+^ T cells tested positive for IL-13 (Figure 5). However, CD4^−^ and IL-13^+^ cells were also present (Figure 8), suggesting that IL-13 was produced by cells other than CD4^+^ cells.

Normally, inflammation increases the concentrations of cytokines such as IL-1β and TNF-α, which increases COX-2; when COX-2 increases, PGE2 increases, which in turn causes feedback that further increases the presence of inflammatory cytokines. In normal analgesia, when COX-2 is inhibited, PGE2 is suppressed, and inflammatory cytokines are inhibited. On the other hand, in NSAID-induced ulcers, COX-1 is also inhibited at the same time, and PGE2 is suppressed; thus, the protective mechanism fails to function, and other external factors cause ulcers [4]. Although other groups have reported on IL-13 and COX, such as reports of IL-13 stimulating the COX pathway [32], to the best of our knowledge, this is the first study to show that IL-13 is located downstream of the COX pathway. If this is the case, it is possible that this signaling pathway is regulated by positive or negative feedback from IL-13 downstream of COX. If this cycle is clarified, it will lead to further elucidation of the COX pathway. However, it is still unclear at this point whether the relationship between the COX pathway and IL-13 is direct or indirect in this experimental system. In the future, we plan to conduct further experiments with COX-2 selective inhibitors.

This study indicates that IL-13 is induced in IDM-induced small intestinal mucosal injury with ulceration, suggesting that there may be a different mechanism underlying the pathogenesis of IDM-induced small intestinal mucosal injury than that of gastric ulcer. As shown in Figure 9, the fact that IL-13 induces cytotoxicity may be a similar mechanism to the cytotoxicity we previously reported in our irradiation model [19]. On the other hand, IL-13-induced cytotoxicity is enhanced in the presence of IDM suggests that IL-13 lies downstream of IDM action (as mentioned earlier); however, corroboration will only be provided by future investigations.

IL-13 transduces its signals via the IL-13 receptor, which consists of the dimers, IL-13Rα1 and IL-4α, and is mainly expressed on epithelial cells [33]. There is also a decoy receptor for IL-13, IL-13Rα2, which has recently been reported to transmit IL-13 signaling without blocking it [34]. Whether or not these receptors and their signaling are involved in this pathology will only be ascertained via future investigations.

There are various reports stating that IL-13 affects the intestinal microbiota in animal models (similar to the present study) [18], that IL-13 production is induced by bacterial and parasitic infections [35], that IL-13 promotes intestinal cell turnover [36], and that NSAIDs may injure the intestine through a TLR4-dependent pathway [37]. Therefore, it is necessary to investigate the possibility that bacteria invaded the ulcerated epithelial barrier and elevated IL-13 levels as a result.

Based on the results of this study, we assert that, if IL-13 (whose levels were elevated in this case) is considered to be an injury factor, we should consider suppressing IL-13 as a healing strategy. The combination of IL-13 inhibitors as well as the administration of NSAIDs can be considered an example of this.

## 4. Materials and Methods

### 4.1. Overall Experimental Design and Evaluation Methods

The mucosal injury murine model was prepared using NSAIDs. First, the degree of injury caused by the systemic effects of the drug was evaluated by biological observation. We assessed the damage (mucosal injury) using pathological analyses, gene expression analyses, immunohistochemistry, flow cytometry, and in vitro experiments in intestinal tissues. All of the experiments were performed per the Institutional Guidelines for the Care and Use of Laboratory Animals in Research and were approved by the local ethics committee of Kitasato University (approval number: 21-06-2, approval date: 4 April 2022, approver: Hidero Kitasato; Chairperson of School of Allied Health Sciences, Kitasato University).

### 4.2. Mice and Drug Treatment

We obtained six-week-old male BALB/cAJcl mice (CLEA-Japan, Tokyo, Japan) and allocated 12–70 of them to each group depending on the experimental systems shown in Figure Legend. All the animals were maintained at 23 °C ± 3 °C in a twelve-hour light–dark cycle. All the mice were bred under specific pathogen-free conditions at the School of Allied Health Sciences, Kitasato University, Japan. The mice were provided with a commercial diet (CRF-1, Oriental Yeast Co., Ltd., Tokyo, Japan) and water ad libitum. All the experiments were performed according to the Institutional Guidelines for the Care and Use of Laboratory Animals in Research and were approved by the local ethics committee at Kitasato University. Indomethacin (IDM) (Merck & Co., Inc., Darmstadt, Germany) was administered via subcutaneous injection once (30 mg/kg) after being dissolved in 0.03 M NaOH, neutralized with 0.05 M HCl, and suspended in saline. Control animals received saline instead of IDM. IDM was administered once on the first day (Day 0). Each mouse’s body weight was measured every other day. All the experimental procedures, such as IDM administration, were conducted by researchers skilled in handling mice. In addition, the biological observation and experimental and data analyses were performed by researchers who had been trained for at least ten years.

### 4.3. Histological Analysis

The tissues were immediately fixed for 24 h in freshly prepared 4% paraformaldehyde in PBS, processed, and paraffin-embedded. Three-micrometer sections were washed in xylene and rehydrated in gradient ethanol aliquots. Sections were stained with hematoxylin-eosin (H&E). An optical microscope was used to observe and photograph the tissues. Tissue images were evaluated using an optical microscope (Olympus Corporation, Tokyo, Japan). The villus height of the jejunum and ileum was measured using H&E images. Paraformaldehyde-fixed, paraffin-embedded sections were heated with 10 mmol/L citric buffer at 121 °C for ten minutes, incubated with 3% H_2_O_2_ /PBS and 0.5%Triton/PBS, blocked with Protein Block (Agilent Technologies, Santa Clara, CA, USA), stained with Goat anti–IL-13 polyclonal antibodies (Santa Cruz Biotechnology, Dallas, TX, USA) and Rabbit anti-goat IgG-HRP antibodies (Santa Cruz Biotechnology), followed by the use of the ImmPACT DAB Substrate Kit (Vector Laboratories, Inc., Newark, CA, USA). The percentage of IL-13-positive areas was calculated using the villus area, with the muscle layer excluded as the denominator. The quantification was performed using Image J (NIH, Bethesda, MD, USA). IL-13 and CD4 double staining were performed using anti-IL-13 IgG-PE and anti-CD4 IgG-FITC antibodies (Thermo Fisher Scientific Inc., Waltham, MA, USA).

### 4.4. Evans Blue Administration and Detection

Evans Blue has a high affinity for serum albumin. Increased vascular permeability leads to the extravasation of albumin-bound Evans Blue dye. Evans Blue (FUJIFILM Wako Pure Chemical Corporation, Osaka, Japan) was injected into the tail vein of each mouse (20 mg/mL in PBS, 100 µg/mouse) and the mouse was left left for 30 min, after which Balb/c mice were anesthetized and perfused with 20–30 mL of PBS. The removed tissue was cut into 5 mm squares. Formamide was added to the minced tissue at a concentration of 0.1 g/mL in order to leak out Evans Blue in the tissues. The absorbance (620 nm) of the leaked Evans Blue was measured using a spectrophotometer (model UV-240, Shimadzu Scientific Instruments, Kyoto, Japan). Dye amounts were calculated from the Evans Blue standard curve (0.5–40 µg/mL).

### 4.5. Real-Time Reverse Transcription-Polymerase Chain Reaction

Transcripts encoding IL–1β, IL–6, TNFα, IFN-γ, IL-4, IL-13, and glyceraldehyde-3-phosphate dehydrogenase (GAPDH) were determined via a real-time reverse transcription polymerase chain reaction (RT–PCR). Briefly, total RNA was purified from tissues using the TRIzol RNA Isolation Reagents (Thermo Fisher Scientific, Waltham, MA, USA). Single-stranded cDNA was generated from the total RNA by reverse transcription using the PrimeScript RT reagent Kit (TAKARA BIO INC., Shiga, Japan), as per the manufacturer’s instructions. Quantitative PCR amplification was performed with SYBR Select Master Mix (Thermo Fisher Scientific, Waltham, MA, USA). The gene-specific primers used were as follows: IL–1β, 5′–gggctgcttccaaacctttg and 5′–aagacacaggtagctgccac; IL–6, 5′–agttgccttcttgggactga, and 5′–tccacgatttcccagagaac; TNFα, 5′–gacgtggaactggcagaaga and 5′–actgatgagagggaggccat; IFN-γ, 5′–tgggtcctgtagatggcattgc, and 5′–tttccgcttcctgaggctggat; IL-4, 5′–atcatcggcattttgaacgaggtc, and 5′–accttggaagccctacagacga; IL-13, 5′–aacggcagcatggtatggagtg, and 5′–tgggtcctgtagatggcattgc; GAPDH, 5′–tgatgggtgtgaaccacgag, and 5′–agtgatggcatggactgtgg. Data were normalized to the level of GAPDH in each sample.

### 4.6. Laser Microdissection (LMD)

A 20 μm thick section was prepared from the frozen tissues and attached to a slide glass. After H&E staining, the ulcer, near ulcer, and morphologically preserved parts were excised using LMD (Thermo Fisher Scientific), and total RNA was extracted from each sample.

### 4.7. Lamina Propria Mononuclear Cell Separation and Primary Cultures

Samples of the total small intestine were rinsed in PBS and then treated with 2 mmol/L EDTA in PBS for 30 min to remove the epithelial cells. The residue was then digested with Sigma Blend Collagenase Type F (Sigma-Aldrich Co, Ltd., St Louis, MO) for 20 min and separated in the Percoll gradient to obtain LPMCs. In the primary cultures, 1 × 106 cells of LPMCs per well were cultured in 100 μL of 10% FCS/RPMI1640 for 24 h. At that time, 40 µL each of PMA (Phorbol 12-Myristate 13 Acetate) (Merck KGaA, Darmstadt, Germany) and lonomycin (Merck KGaA) were added to the culture medium. After culturing for 24 h, the supernatant was collected and stored in a freezer at −80 °C until it was used for ELISA. LPMCs were washed with PBS and stored in a freezer at −80 °C until they were used for mRNA expression analyses, ELISA, and flow cytometry.

### 4.8. ELISA

Mouse IL-13 DuoSet ELISA (Bio-Techne Corporation, Minneapolis, MN) was used for IL-13 detection. Briefly, a capture antibody solution (1–10 µg/mL; 100 µL/well) was added to the plate; the surface of the plate was covered with a plastic seal and incubated overnight at 4 °C. The seal was peeled off to remove the capture antibody solution, after which each well was washed twice with 200 µL of PBS. Two hundred microliters of 0.5% BSA/PBS were added to each well as a blocking solution. The plate surface was covered with a plastic seal and incubated overnight at 4 °C. One hundred microliters of the diluted sample solution were added to each well and reacted at 37 °C for 90 min. The liquid was removed, and each well was washed twice with 200 µL of PBS. One hundred microliters of detection antibody solution, diluted with blocking solution, were added to each well. The surface of the plate was covered with a plastic seal, reacted at room temperature for two hours, the liquid was removed, and the plate was washed four times with PBS. One hundred microliters of the substrate/color developer solution were added to each well and reacted. Absorbance was detected using a microplate reader (Bio-Rad Laboratories, Hercules, CA, USA).

### 4.9. Flow Cytometry

Approximately 1 × 10^5^ cells of LPMCs were dissolved in RPMI1640 supplemented with PMA and lonomycin. The mixture was then incubated at 37 °C for four hours. After adding cold PBS to stop the reaction, the mixture was centrifuged at 4 °C, 300× *g*, for five minutes. Labeled antibodies that recognize cell surface antigens (Anti-CD3 IgG-PE, anti-CD4 IgG-FITC, anti-B220 IgG-PE, anti-F4/80 IgG-FITC, IgG1 Isotype Control-PE, IgG1 Isotype Control-FITC, IgG2a Isotype Control-PE or IgG2a Isotype Control-FITC antibodies; Thermo Fisher Scientific Inc.) were added. The Intra Prep Reagent 1 (Beckman Coulter Inc., Brea, CA, USA) was added to each sample tube. The product was mixed well and incubated at room temperature in the dark for fifteen minutes. After a fifteen-minute incubation in the dark, a 1 mL aliquot of PBS was asses, the mixture was centrifuged at 300× *g* for five minutes at room temperature, and the supernatant was aspirated off. The cells were well loosened, and Intra Prep Reagent 2 (Beckman Coulter Inc.) was added to 100 µL aliquots, after which the resultant solution was mixed gently and incubated at room temperature in the dark for five minutes. Anti-IL-13 IgG-PE antibody (Thermo Fisher Scientific Inc.) was added and mixed gently. After a fifteen-minute incubation at room temperature in the dark, PBS was added, the resultant mixture was centrifuged at 300× *g* for five minutes at room temperature, and then the supernatant was aspirated off. Cells lysed with PBS were analyzed using a flow cytometer (Becton, Dickinson and Company, Franklin Lakes, NJ, USA).

### 4.10. Cell Culture and Cell Viability Assay

Cell cultures of Caco-2 cells obtained from human colonic carcinoma were performed in a culture medium comprising DMEM with 4 mmol/L glutamine and 10% FBS in a 10 cm^2^ tissue culture dish. The cells were maintained at 37 °C in an atmosphere of 5% CO_2_/95% air in a CO_2_ incubator. The cells were sub-cultured via partial digestion with 0.25% trypsin and 1 mmol/l EDTA in a PBS solution. The Caco-2 cells (passages 2–4) were plated on a 96-well culture plate (1 × 10^5^ cells/well). After 24 h, IDM and/or recombinant human IL-13 (Thermo Fisher Scientific, Waltham, MA, USA) was/were added to the culture supernatant (0, 50, or 100 µg/mL final concentration) and cultured for 12–96 h. The viability of Caco-2 cells was determined via a cell proliferation assay using the MTT reagent (Nacalai Tesque, Inc., Tokyo, Japan). An MTT solution was added to each well. Following a three-hour incubation at 37 °C, a microplate reader (Bio-Rad Laboratories, Hercules, CA) was used to determine the absorbance at 570 nm (reference at 650 nm). The cell viability percentage was calculated based on the absorbance measured relative to that of cells not exposed to the MTT solution.

### 4.11. Statistics

Quantitative data were expressed as mean values and standard deviations (mean ± SD), whereas qualitative data were expressed as frequencies and percentages. Statistical analyses were performed using GraphPad Prism version 6.0 (GraphpPad Software, Inc., San Diego, CA, USA). Multiple comparison tests were performed using the two-way analysis of variance with Tukey’s post hoc test. A comparison of the averages of the two groups was performed using the *t* test. The normality of the distribution of continuous variables was evaluated using the Kolmogorov–Smirnov test. Most data were confirmed using the post hoc sample size power calculation to verify whether there was adequate power to evaluate the outcomes of studies.

## 5. Conclusions

The findings of this study indicate that IL-13 is closely involved in the pathogenesis of NSAID-induced small intestinal mucosal injury and is expected to contribute to the development of anti-inflammatory drugs with fewer side effects.

## Figures and Tables

**Figure 1 ijms-24-14971-f001:**
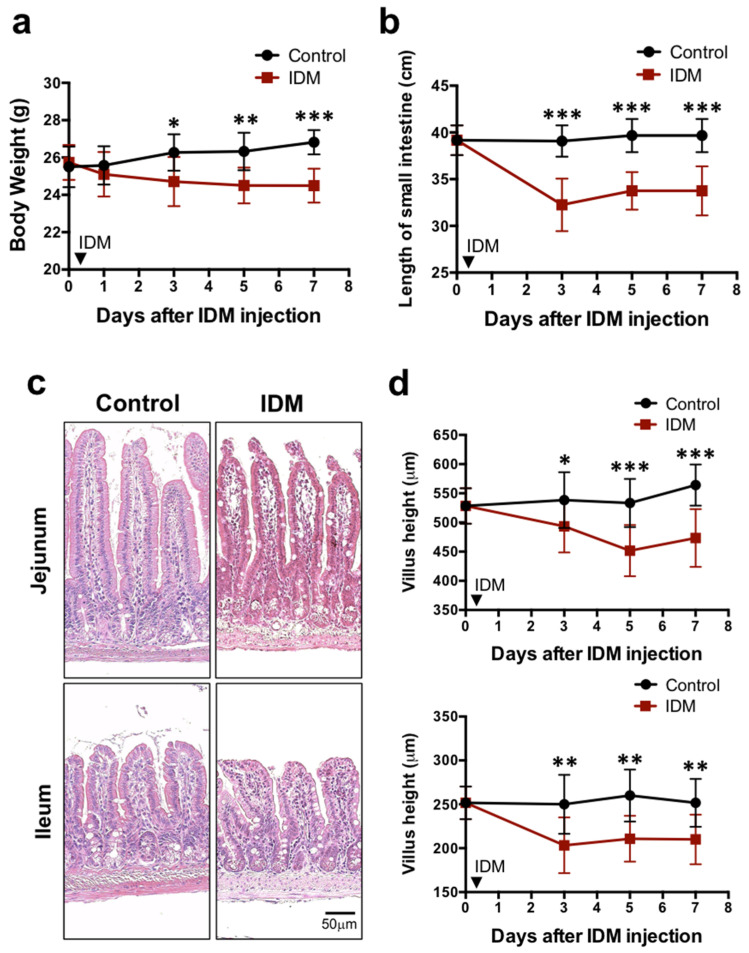
Biological and histological changes in the indomethacin (IDM)-induced intestinal mucosal injury model. Seventy-two mice (36 mice each for the control and IDM groups) were used. The IDM group was administered a single dose of IDM by subcutaneous injection. From day 1 of IDM administration (which was defined as day 0), body weight (**a**), the length of the small intestine (**b**), and the villus height (**c**,**d**) were measured. Statistical analyses were performed using two-way analysis of variance with Tukey’s post hoc test. * *p* < 0.05, ** *p* < 0.01, *** *p* < 0.001.

**Figure 2 ijms-24-14971-f002:**
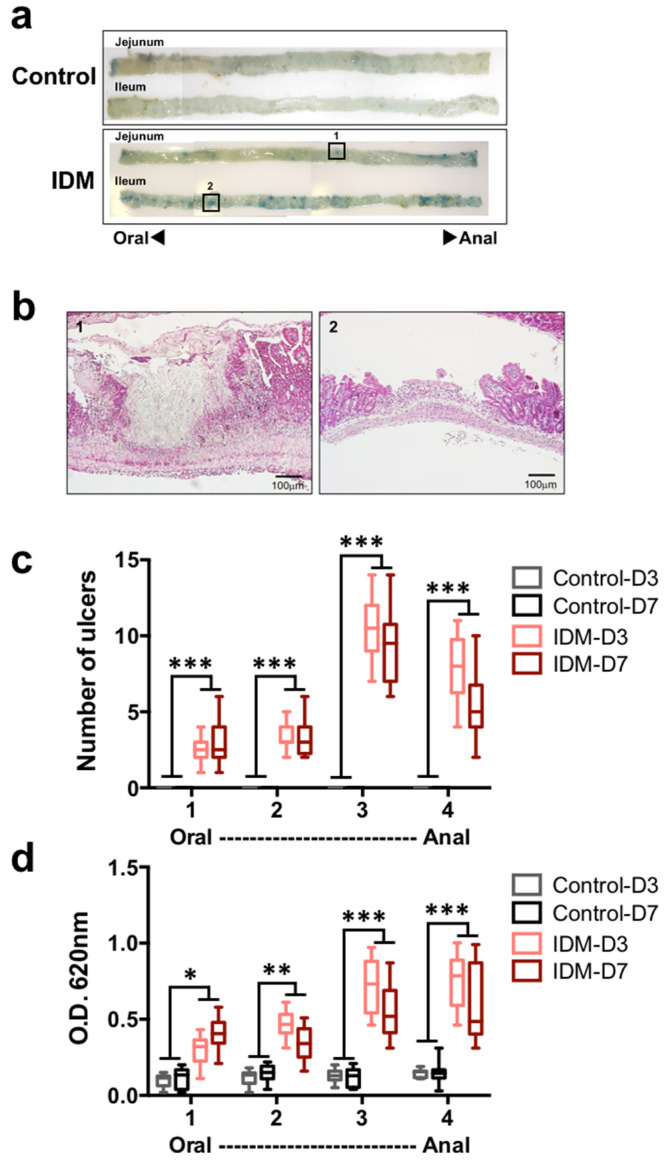
Ulceration in the small intestine of the indomethacin (IDM)-induced intestinal mucosal injury model. Forty-eight mice (12 mice for each point) were used. The control and indomethacin (IDM) groups were tail-injected with Evans Blue and left for 30 min. After perfusion with PBS, the small intestine was harvested and observed. The blue spots were inflammatory sites with increased vascular permeability (**a**). The ulcer tissues observed in (**a**) -1 or -2 were stained with H&E (**b**). The ulcers (**c**) were counted and the extravasation of Evans blue in the tissue (**d**) was measured via absorbance. Statistical analyses were performed using the two-way analysis of variance with Tukey’s post hoc test. * *p* < 0.05, ** *p* < 0.01, *** *p* < 0.001.

**Figure 3 ijms-24-14971-f003:**
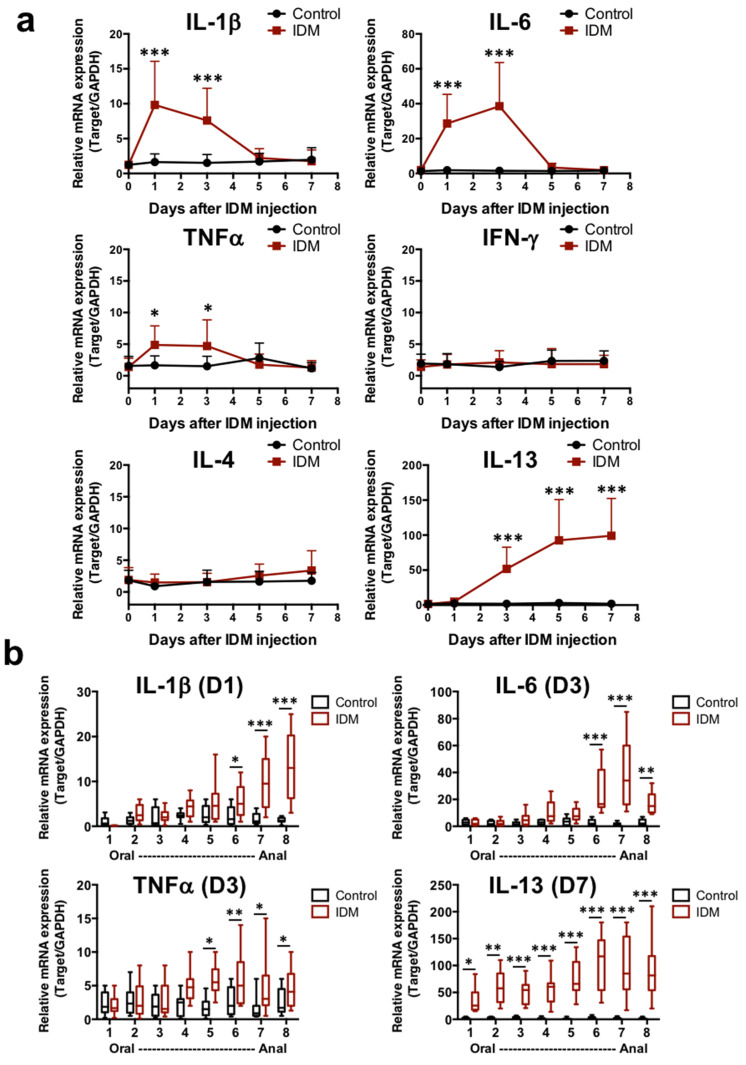
mRNA expression analysis of cytokines in small intestinal tissues of the indomethacin (IDM)-induced intestinal mucosal injury model. Seventy mice (seven mice for each point) were used. mRNA expressions of several cytokines in ileal tissues were sequentially analyzed via real-time PCR (**a**). From the analysis results of Figure 3a, the expression differences were confirmed for each small intestine site on days when mRNA expressions were significantly increased (**b**). Statistical analyses were performed using the two-way analysis of variance with Tukey’s post hoc test. * *p* < 0.05, ** *p* < 0.01, *** *p* < 0.001.

**Figure 4 ijms-24-14971-f004:**
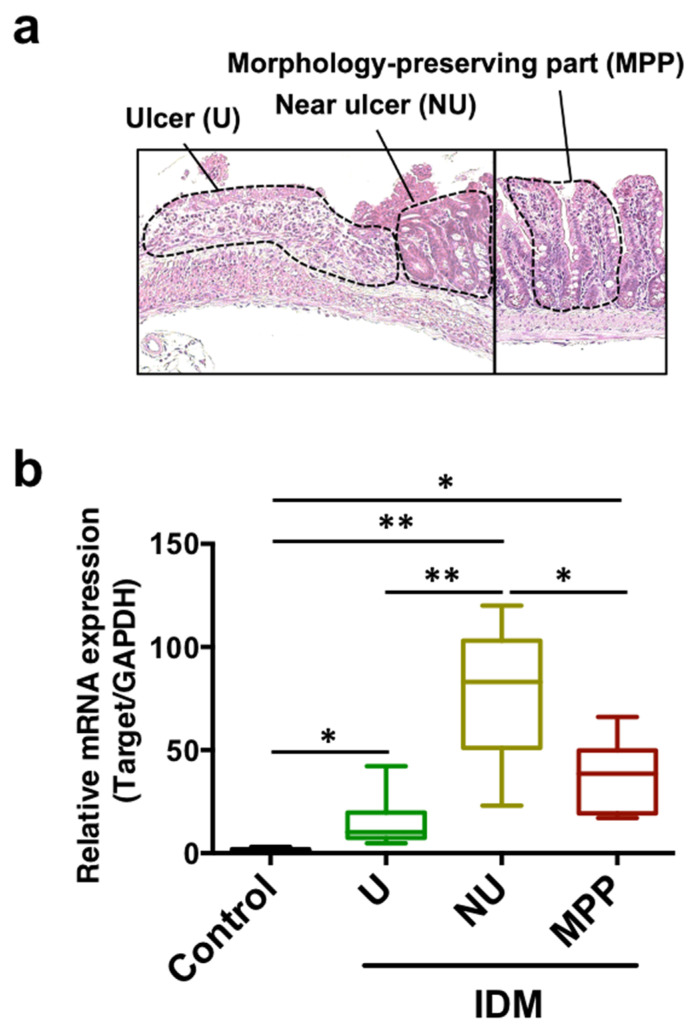
mRNA expression analysis of IL-13 in small intestinal tissues divided by parts of the indomethacin (IDM)-induced intestinal mucosal injury model. Sixteen mice (eight mice for each point) were used on day 7 after IDM administration. The small intestinal tissue was separated into each site, namely, the ulcer (U), near ulcer (NU), and the morphologically preserved part (MPP), via the laser microdissection method (**a**). mRNA expressions of IL-13 were analyzed using real-time PCR (**b**). Statistical analyses were performed using the two-way analysis of variance with Tukey’s post hoc test. * *p* < 0.05, ** *p* < 0.01.

**Figure 5 ijms-24-14971-f005:**
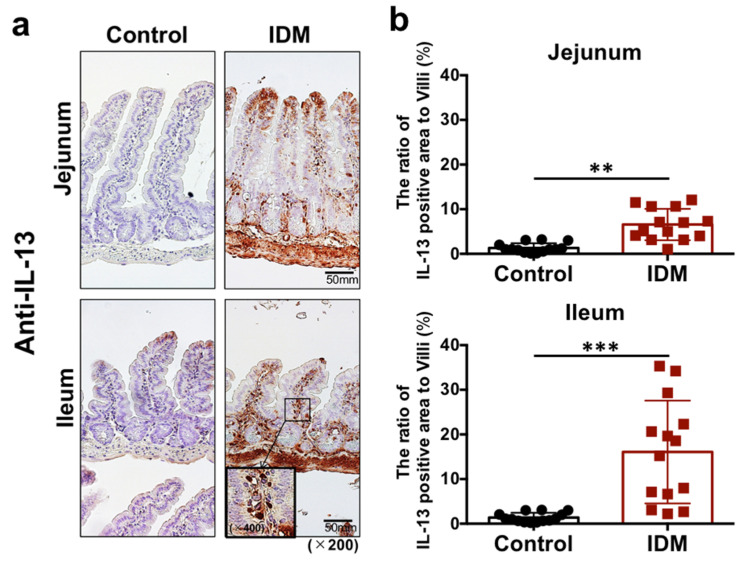
IL-13 detection in the small intestine of the indomethacin (IDM)-induced intestinal mucosal injury model. Twenty-eight mice (14 mice each for the control and IDM groups) were used. Jejunal and ileal tissues were determined using IL-13 immunohistochemistry (**a**). Percentages of IL-13-positive areas per villi were calculated (**b**). Statistical analyses were performed using the *t* test. ** *p* < 0.01, *** *p* < 0.001.

**Figure 6 ijms-24-14971-f006:**
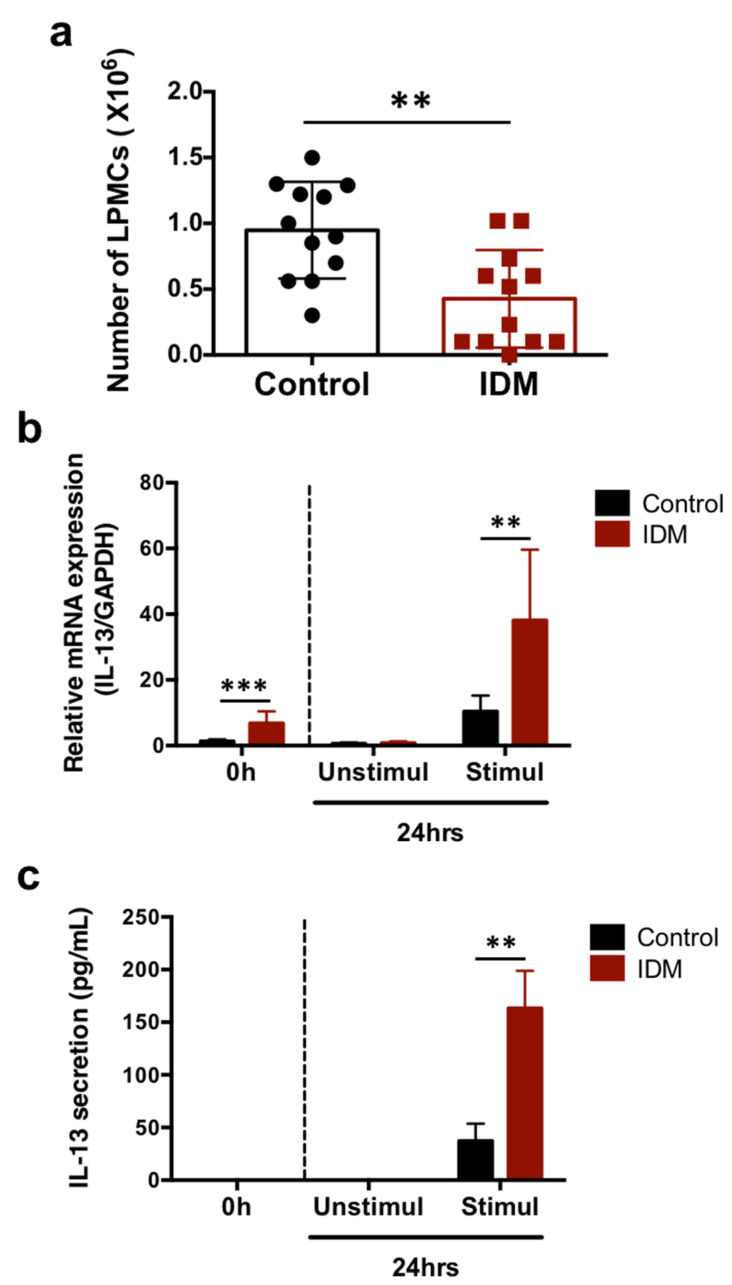
IL-13 mRNA expression and secretion in the LPMCs of the small intestine in the indomethacin (IDM)-induced intestinal mucosal injury model. Twenty-four mice (12 mice each for the control and IDM groups) were used. The separated lamina propria mononuclear cells (LPMCs) were stimulated with PMA and ionomycin for 24 h. The LPMCs were counted (**a**). mRNA expressions of IL-13 in LPMCs were analyzed via real-time PCR, while IL-13 secretions in the culture supernatant of LPMCs were analyzed using ELISA (**b**,**c**). Statistical analyses were performed using the *t* test (**a**) and two-way analysis of variance (ANOVA) with Tukey’s post hoc test (**b**,**c**). ** *p* < 0.01, *** *p* < 0.001.

**Figure 7 ijms-24-14971-f007:**
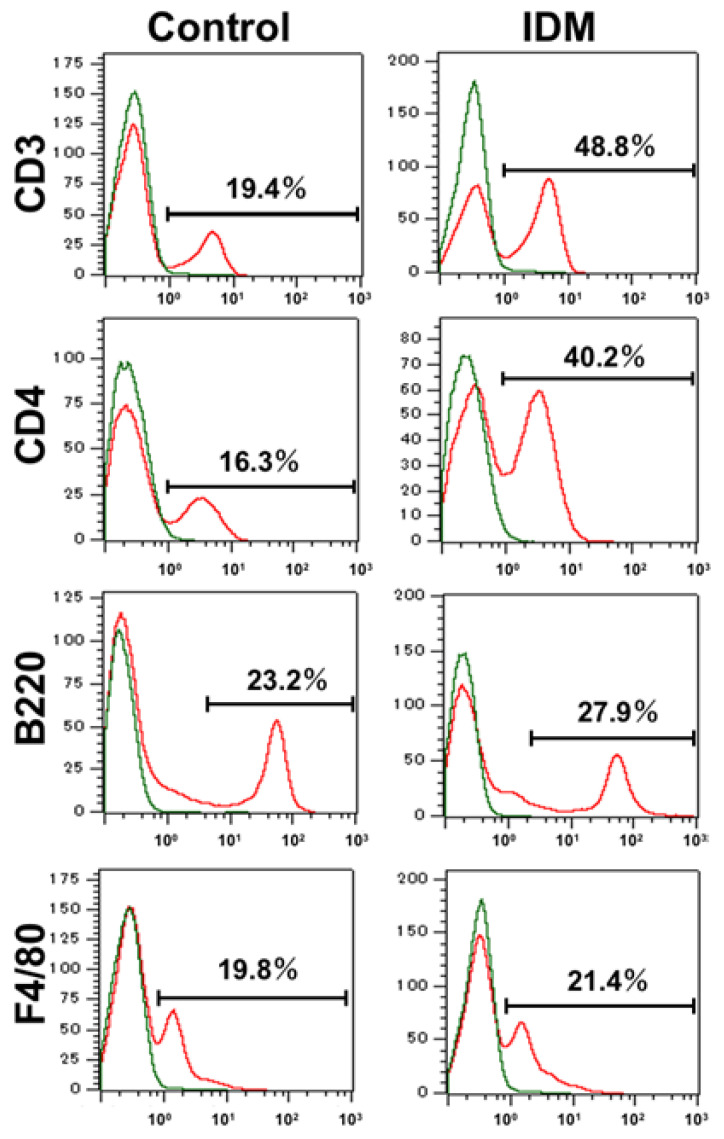
Identification and characterization of the cell subpopulations of LPMCs of small intestine in indomethacin (IDM)-induced intestinal mucosal injury model. Twenty-four mice (12 mice each for the control and IDM groups) were used. Cell type identification was performed on stimulated LPMCs via flow cytometry (FACS).

**Figure 8 ijms-24-14971-f008:**
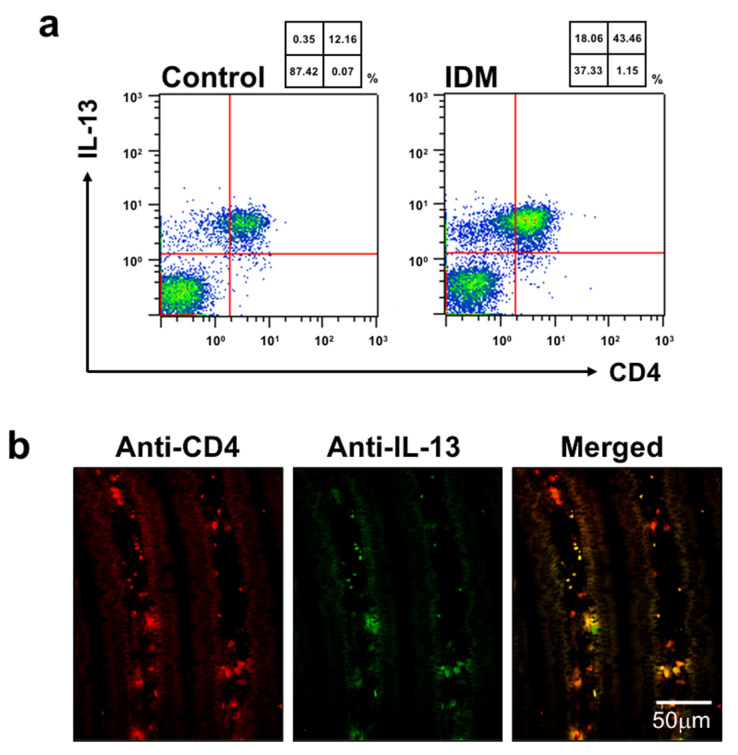
Identification of cells expressing IL-13 in the small intestine of the indomethacin (IDM)-induced intestinal mucosal injury model. Twenty-four mice (12 mice each for the control and IDM groups) were used. CD4-positive and IL-13-positive cells were detected (**a**). The CD4-positive and IL-13-positive cells in jejunal tissues were determined via CD4 and IL-13 double staining (**b**).

**Figure 9 ijms-24-14971-f009:**
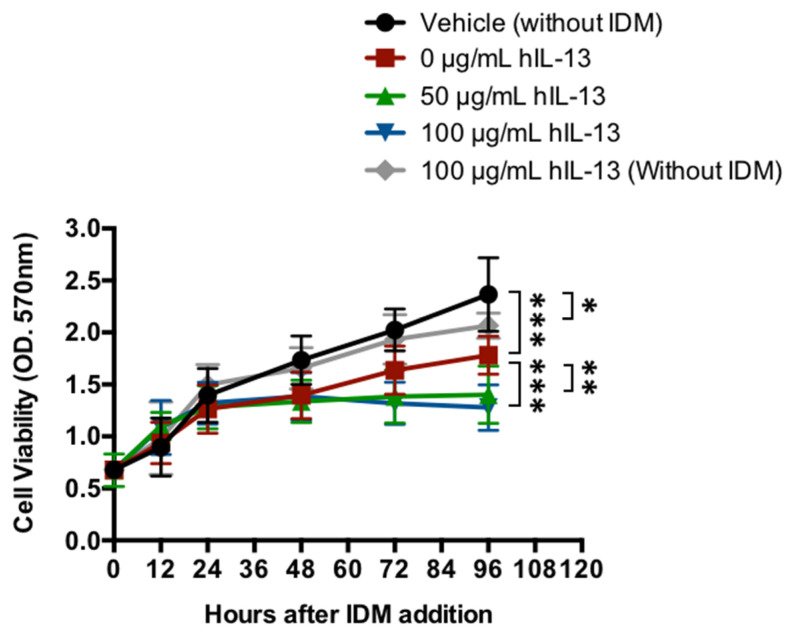
Cytotoxicity of IL-13 in intestinal epithelial cells. The cell viability of the human colonic carcinoma cell line (Caco-2 cells) added to recombinant IL-13 proteins in the presence of IDM was evaluated. Statistical analyses were performed using the two-way analysis of variance (ANOVA) with Tukey’s post hoc test. * *p* < 0.05, ** *p* < 0.01, *** *p* < 0.001.

## Data Availability

All datasets generated for this study are included in the manuscript. The raw data presented in this study are available on request from the corresponding author.

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
