# Peer review of "Interleukin-13 Mediates Non-Steroidal Anti-Inflammatory-Drug-Induced Small Intestinal Mucosal Injury with Ulceration"

_ijms, 2023, doi:10.3390/ijms241914971_

Round 1
Reviewer 1 Report
In the present experimental study performed on a murine model of indomethacin (IDM) induced small intestinal injury, Kawashima et al found that overexpression of IL13 is a prolonged event occurring in the mucosa and CD4-T cells in the lamina propria. Main comments:
1) Page 1 line 19: abdominal pain cannot be considered as a side effect of NSAIDs.
2) Does the analysis reported in fig. 5 refer only to epithelial cells? Authors could perform the same analysis for cells in the lamina propria as well, considering the results of the experiment described in paragraph 2.7.
3) What is PMA? Why should it have stimulatory properties on IL13?
4) Figure 3b shows a progressive gradient in proinflammatory cytokines expression. This finding should be better discussed.
5) Authors did not evaluate whether small bowel injury paralleled with worsening of mice clinical conditions, for example by investigating the number of bowel movements or presence of bloody stools.
Reviewer 2 Report
Review of manuscript entitled „Interleukin-13 mediates non-steroidal anti-inflammatory drug-induced small intestinal mucosal injury with ulceration”
The work presented on mice model effect of IL-13 and indomethacin on injury of small intestinal mucose. Exactly was described eight days obsevations of the pathophysiological mechanism involved in that dynamic process. Changes of many biological, histological and immunological parameters in gene and protein level, as well as, cells type identification were estimated. These many investigations, which let them found that IL-13 involve in small intestinal mucosal injury and indomethacin take a part in stimulation of its secretion.
The work contain shortcomings, as follows
1) Abbreviations should be explained the first time they are mentioned.
The manuscript is sometimes not easy to read when the explanations of abbreviations we can find only at the end of the text (in "Materials and Methods" chapter). For example: You use the term FACS in "Results" without explanation, as more others in the text.
2) Figure 9 is not readable.
3) Why did you not shown these results (line 86 as a numer of citation and 289 as a next drawing e.g. e)
4) Next the flow cytometry (in "Materials and Methods" chapter) and fluorescence activated cell sorting (FACS in Results) you use interchangeably.There are some key differences that distinguish the two processes (I guess this needs to be verified).
5) The "Statistics" chapter is poorly described. See on the descriptions described in the "Results" chapter.
Round 2
Reviewer 1 Report
Answers were fine
none